# Concussion/Mild Traumatic Brain Injury (TBI) Induces Brain Insulin Resistance: A Positron Emission Tomography (PET) Scanning Study

**DOI:** 10.3390/ijms22169005

**Published:** 2021-08-20

**Authors:** Sathiya Sekar, Raja Solomon Viswas, Hajar Miranzadeh Mahabadi, Elahe Alizadeh, Humphrey Fonge, Changiz Taghibiglou

**Affiliations:** 1Department of Anatomy, Physiology and Pharmacology, College of Medicine, University of Saskatchewan, 107 Wiggins Road, Saskatoon, SK S7N 5E5, Canada; sathiya.sekar@usask.ca (S.S.); h.miranzadeh@usask.ca (H.M.M.); 2Department of Medical Imaging, College of Medicine, University of Saskatchewan, Saskatoon, SK S7N 0W8, Canada; rav137@mail.usask.ca (R.S.V.); elahe.alizadeh@usask.ca (E.A.); 3Department of Medical Imaging, Royal University Hospital (RUH), Saskatoon, SK S7N 0W8, Canada

**Keywords:** cellular prion protein, flotillin-1, CAP, brain insulin resistance, mild traumatic brain injury, motor and cognitive functions

## Abstract

Brain injury/concussion is a growing epidemic throughout the world. Although evidence supports association between traumatic brain injury (TBI) and disturbance in brain glucose metabolism, the underlying molecular mechanisms are not well established. Previously, we reported the release of cellular prion protein (PrPc) from the brain to circulation following TBI. The PrPc level was also found to be decreased in insulin-resistant rat brains. In the present study, we investigated the molecular link between PrPc and brain insulin resistance in a single and repeated mild TBI-induced mouse model. Mild TBI was induced in mice by dropping a weight (~95 g at 1 m high) on the right side of the head. The procedure was performed once and thrice (once daily) for single (SI) and repeated induction (RI), respectively. Micro PET/CT imaging revealed that RI mice showed significant reduction in cortical, hippocampal and cerebellum glucose uptake compared to SI and control. Mice that received RI also showed significant motor and cognitive deficits. In co-immunoprecipitation, the interaction between PrPc, flotillin and Cbl-associated protein (CAP) observed in the control mice brains was disrupted by RI. Lipid raft isolation showed decreased levels of PrPc, flotillin and CAP in the RI mice brains. Based on observation, it is clear that PrPc has an interaction with CAP and the dislodgment of PrPc from cell membranes may lead to brain insulin resistance in a mild TBI mouse model. The present study generated a new insight into the pathogenesis of brain injury, which may result in the development of novel therapy.

## 1. Introduction

Traumatic brain injury (TBI) is a growing epidemic throughout the world with global incidences of 69 million individuals every year [1,2]. However, the occurrences of TBI have not been well documented as patients are not hospitalized due to mild injury. TBI has been increasingly accepted as one of the major external risk factors in the development/progression of neurodegenerative diseases [1,3,4,5,6,7,8]. A positive association between type 2 diabetes, insulin resistance (IR) and neurodegenerative diseases such as Alzheimer’s and Parkinson’s diseases has been demonstrated [9,10,11,12,13,14,15,16,17]. The impaired brain insulin signaling leads to neuronal dysfunction and increases neurodegeneration. 

Studies showed the dysregulation of insulin-dependent glucose utilization in the brain links to cognitive deficits and neurodegeneration in TBI [18,19]. Further, insulin sensitizing treatment led to the suppression of damage caused by the TBI and promoted recovery from the insult [20,21]. Positron emission tomography (PET) imaging using ^18^F-FDG tracer was the most widely used technique to non-invasively measure any changes in glucose uptake in TBI patients and animal models [22,23,24,25]. The sensitive and quantitative properties of PET with high affinity make it the method of choice in the development of various therapeutic applications. Although considerable evidence in animal models of mild TBI and human subjects succumbing to sport concussions or veterans exposed to blast-induced mild TBI suggests dysregulation in brain glucose metabolism, the underlying molecular mechanisms are not well established [26,27,28]. 

Taghibiglou’s team has reported the release of lipid rafts of localized cellular prion protein (PrPc) from the brain to circulation following sport concussion and blast-induced brain injury, which may serve as a biomarker for concussion/mild TBI [29,30]. The PrPc level was found to be decreased in hippocampal and cortical slices of IR rats [31]. In addition, the PrPc level was found to be increased in the circulation and decreased in cortical and hippocampal slices of repeated mild TBI mice brains [32]. Based on these observations, we hypothesize that PrPc may play an important role in central insulin sensitivity and brain glucose uptake/metabolism. Repeated mild TBI causes brain IR and dysregulation of glucose metabolism; partly this may be due to the dislodgment of PrPc. An interaction between the lipid raft localized flotillin and CAP (a key marker in the secondary insulin signaling pathway) has been demonstrated [33,34], and flotillin also interacts with PrPc [35,36]. In the present study, we investigated the molecular link between PrPc and brain insulin resistance (CAP/Cbl/TC10 pathway) in a single and repeated mild TBI-induced mouse model. As evidence supports the persistent decrease in brain glucose metabolism in repeated mild TBI patients [28], the study was undertaken to clearly elucidate the molecular link between them and whether the single induction or repeated induction generates evident impairment in mTBI.

## 2. Results

### 2.1. Determination of Glucose Uptake in TBI Mice Brains Following Single and Repeated Induction

To study glucose uptake in TBI mice brains, mice received ^18^F-FDG via a tail vein following single or repeated induction followed by dynamic micro PET/CT imaging of the head for 30 min. Micro PET/CT imaging was performed on day 0 (before the TBI induction), and at 24 h, and 7, 14, 21 and 28 days post induction. Representative PET images of the controls and mice with single and repeated induction are shown (Figure 1). A distinct reduction of ^18^F-FDG uptake/staining was observed in TBI mouse brain following single induction on day 1, compared with the control mice brains. Subsequent scanning of the mice brains once weekly showed that glucose uptake slowly recovered after day 7 for the single induction mice, while those with repeated induction showed persistent reduction in glucose uptake until day 28 post induction compared with control mice brains. 

Quantification of glucose uptake was represented as % injected activity per gram (%IA/cc) (Figure 2). At day 0, all ^18^F-FDG uptake was statistically (*p* > 0.05) the same in all three groups. Compared with control on day 7, those with single induction showed a significant decrease (*p* < 0.05) in ^18^F-FDG uptake in the hippocampus and whole brain. ^18^F-FDG uptake in the single induction mice showed a trend of returning to normal levels from day 14. Repeated induction showed persistent reduction in ^18^F-FDG uptake and the values were found to be statistically significant in the hippocampal region on days 21 and 28, compared to the control mice (*p* < 0.05) and their respective day 0 animals (*p* < 0.01 and 0.05, respectively). 

The percentage decrease of ^18^F-FDG uptake in whole brain, cortex, cerebellum and hippocampus was found to be 36.35, 38.94, 37.48 and 38.48%, respectively, on day 1; 51.08, 54.22, 42.49 and 49.90%, respectively, on day 7; 33.55, 39.18, 18.79 and 37.85%, respectively on day 14; 29.54, 32.58, 24.19 and 29.00%, respectively, on day 21; and 36.72, 35.71, 30.77 and 37.44%, respectively, on day 28. 

In repeated induction, TBI mice brains showed a significant decrease in ^18^F-FDG utilization on days 21 (*p* < 0.05 in cortex and hippocampus regions) and 28 (*p* < 0.01 in cortex and *p* < 0.05 in cerebellum, hippocampus and striatal regions), in comparison to the control mice brains. In addition, a significant decrease (*p* < 0.05) in ^18^F-FDG uptake was observed in whole brain and hippocampal regions, compared to their respective day 0 imaging. The percentage decrease of glucose uptake was found to be 28.92, 30.43, 30.94 and 35.77%, respectively, on day 1; 29.04, 32.53, 18.74 and 27.42%, respectively, on day 7; 34.23, 16.71 and 38.17%, respectively, on day 14; 31.52, 35.14, 41.56 and 53.64%, respectively, on day 21; and 39.90, 34.59, 50.72 and 48.38%, respectively, on day 28. The images showed that repeated TBI induction resulted in a permanent decrease in glucose metabolism in mice brains and a clear significance level may be attained with increased animal numbers.

### 2.2. Determination of Functional Outcomes in TBI Mice following Single and Repeated Induction

#### 2.2.1. Rota Rod Test

On the 28th day post-TBI induction, the mice were assessed for motor coordination. In rota rod tests, the time spent by the mice on the rotating rod was found to be significantly decreased (*p* < 0.01) for those with repeated induction, while a non-significant reduction was observed in single induction mice compared to the control mice (Figure 3A). 

#### 2.2.2. Beam Walk Test

The beam walk test showed a significant increase (*p* < 0.05) in the time taken to cross the narrow beam in single and repeated induction mice compared to the control mice. Immobility periods and number of foot slips were significantly increased (*p* < 0.05 and 0.01, respectively) in those with repeated induction, while a non-significant increase was observed in single induction mice compared to the control mice (Figure 3B–D). 

#### 2.2.3. Open Field Exploratory Test

In the open field test, a significant increase in immobility periods (*p* < 0.01 and 0.001, respectively) was observed in single and repeated induction mice in comparison to the control mice. The number of total and center squares crossed by the single and repeatedly induced TBI mice was found to be non-significant compared to the control mice (Figure 3E–G).

#### 2.2.4. Novel Location Recognition Test

Novel location recognition tests were conducted to assess the cognitive function of the TBI mice. The tests revealed that the mice that underwent single and repeated induction spent less time in the novel location in comparison to the control mice and the values were found to be significant (*p* < 0.01 and 0.001, respectively) (Figure 3H).

### 2.3. Measurement of Plasma PrPc Level in TBI Mice following Single and Repeated Induction

In order to analyze whether the dislodgment of PrPc reaches the systemic circulation, we measured PrPc levels in plasma, 24 h after the last TBI induction. TBI mice that underwent repeated induction showed a significant increase (*p* < 0.01) in plasma PrPc level when compared with single induction and control mice, while a non-significant increase was observed in single induction compared with control mice (Figure 4).

### 2.4. Validation of PrPc, Flotillin, CAP, APS and TC10 Proteins in Lipid Raft Region of TBI Mice Brains Following Single and Repeated Induction

For further behavioral assessment, the mice were sacrificed and their brains excised out. The lipid raft regions from the brains were isolated by sucrose gradient centrifugation method. Twelve fractions were separated in each sample and blotted for PrPc, flotillin, CAP, APS and TC10 proteins. The curdy white precipitate was collected as the 4th fraction that was considered to be the lipid raft fraction. The results obtained from the present study showed that these proteins were markedly decreased in single induction mice compared with control mice brains, while no expression was observed in repeated induction mice similar to PrPc null mice brains (Figure 5).

### 2.5. Interaction between the PrPc, Flotillin and CAP in TBI Mice Brains Following Single and Repeated Induction

In order to determine any possible alteration in the interaction between the PrPc, flotillin and CAP, the control, TBI and PrPc null mice brains were immunoprecipitated for the abovementioned proteins and immunoblotted with interactive proteins. A marked decrease in CAP and flotillin were observed in single and repeated induction mice when immunoprecipitated with PrPc. Similarly, decreased flotillin levels were observed when immunoprecipitated with CAP and decreased PrPc and CAP levels were observed in flotillin immunoprecipitation. The PrPc null mice brains were used as a positive control for comparing the expressions with single and repeated induction mice (Figure 6).

## 3. Discussion

The present study is the first evidence for the existence of PrPc as a molecular link between TBI and insulin resistance. Glucose uptake measured using micro PET/CT scanning was markedly reduced from day 1, which was partly reversed in single induction mice brains, and persistent reduction was observed in those with repeated induction. Dislodgement of PrPc from the brain to circulation was observed in TBI mice. Motor and cognitive deficits were observed in both single and repeated induction mice. Lipid raft isolation and co-immunoprecipitation experiments revealed the disruption of interaction between PrPc/flotillin/CAP in single and repeated induction mice. Based on the above observation, we suggest that PrPc plays a main role in the existence of insulin resistance by disturbing CAP/Cbl/TC10 signaling cascades (PI3K-independent glucose uptake) following TBI.

Hyperglycemia was found to be one of the most common secondary complications in TBI [37,38]. Disruption of cellular uptake of glucose results in metabolic energy demand to the neuronal cells following TBI [39]. Evidence has shown that the changes in glucose uptake and its metabolism after TBI leads to neurodegenerative diseases including Alzheimer’s disease [39,40]. O’Connell and his team showed an association existed between brain glucose metabolism and TBI in clinical patients [41]. Our results from the present study were found to corroborate O’Connell’s findings. ^18^F-FDG uptake in single induction mice was reduced significantly 24 h after injury and slowly restored from day 14, while repeated induction mice showed persistent reduction of ^18^F-FDG uptake in various brain regions. This clearly showed that decreased glucose utilization by various brain regions is associated with a hyperglycemic state in TBI. This persistent decrease in glucose utilization and increased hyperglycemic state in TBI may be one of the causative factors for the existence of long-term neurodegenerative diseases. 

It is well known that PrPc plays a vital role in the central and peripheral nervous system and misfolding of PrPc leads to fatal neurodegenerative diseases. In addition, PrPc is involved in normal cognitive functions and reports showed that PrPc null mice showed memory deficits [42]. We previously reported that the release of PrPc from the brain to circulation following sport concussion and blast-induced brain injury may serve as a biomarker for TBI [29,30,43]. The increased plasma PrPc level observed in the present study was found to corroborate the above findings. Further, the motor and cognitive deficits observed in TBI mice may be due to the dislodgment of PrPc and its subsequent cellular changes. 

PrPc levels were found to be decreased in hippocampal and cortical slices of insulin-resistant rats and repeated TBI mice [30,32]. Evidence supports that the chronic central insulin resistance observed in traumatic brain injury leads to neurodegenerative diseases [18,44,45]. Although various investigations support the link between brain insulin resistance and TBI, the exact molecular link between them has not been well studied. In the present work, we studied the link between PrPc and brain insulin resistance. We showed that PrPc interacts with flotillin and CAP, an insulin signaling protein component. Disruption of this interaction leads to alterations in the CAP/Cbl/TC10 pathway which may be indicative for the discussed insulin resistance in TBI. The lipid raft localized PrPc interacts with flotillin and CAP in the raft region for their subsequent functioning. In the present study, the decreased PrPc level and its interaction with flotillin-1 and CAP in TBI mice brains suggests that PrPc plays a potential molecular linking role in brain insulin resistance in TBI.

In conclusion, repeated TBI mice brains showed consistent and/or prolonged damage to insulin signaling compared to the single TBI mice brains. Further, PrPc appears to play a vital role in insulin resistance following TBI in mice brains. Dislodgement of PrPc following TBI leads to the disruption of interaction between the flotillin and CAP. In addition, the glucose uptake was distinctly reduced in the single and repeated induction mice brains from day 1 following last TBI induction, as evidenced by ^18^F-FDG uptake/metabolism. This suggests that the disruption of insulin signaling cascade may occur through PrPc dislodgement. Thus, PrPc may be considered as the molecular link between TBI and brain insulin resistance. The possible role of PrPc in the CAP/Cbl/TC10 signaling cascade is illustrated in Figure 7. Further studies are warranted with increased animal numbers to clearly elucidate this mechanism in different brain regions, and replacing PrPc in the lipid raft may halt the initiation and/or existence of insulin resistance and long-term neurodegenerative diseases following TBI. Since female animals showed a protective effect in functional outcomes and pathology following TBI [46], male mice were used in the present study. The outcome of the current study may pave the way for future use of agents with potential upregulating effects on PrPc gene and/or anti-diabetic drugs as therapeutic options in the treatment of TBI patients.

## 4. Materials and Methods

### 4.1. Chemicals and Reagents

Fludeoxyglucose F18 (^18^F-FDG) was produced at the Saskatchewan Centre for Cyclotron Sciences (SCCS), University of Saskatchewan, Canada. PrPc ELISA kit was procured from Biomatik Corporation, anti-rabbit APS (#PA5-17608), anti-rabbit (#7074S) and anti-mouse IgG (#7076S) from Cell Signaling Technology, Dynabeads^®^ Protein G (#10003D) from Thermofischer Scientific and anti-rabbit flotillin 1 (ab41927), anti-mouse PrPc (ab61409), and anti-mouse CAP (ab182604) antibodies were obtained from Abcam. All other chemicals and reagents purchased were of analytical grade. 

### 4.2. Animal Husbandry

Seven to nine-weeks-old male C57BL/6 mice (Strain code: 027; Charles River, Montreal, Canada) and PrPc null mice generously provided by Dr. A. Aguzzi (Zurich University, Switzerland) were used for the study. All the animals were housed in groups (2–3 animals/cage), under 12 h light and 12 h dark cycle with temperature controlled (~21 °C) and were provided with standard food and water ad libitum. The protocol was approved by the University Animal Care Committee (UACC; protocol # 20150043), University of Saskatchewan, Saskatoon SK, Canada, and the experiment was conducted following the Guidelines of the Canadian Council on Animal Care (CCAC).

### 4.3. Experimental Design 

C57BL/6 mice were acclimatized to the laboratory conditions for a period of 7 days before experimentation. During acclimatization, the animals were trained in handling procedure. Animals were grouped (*n* = 10) into three as follows: Group I: control (no induction), Group II: TBI (single induction), and Group III: TBI (repeated induction).

Animals were anesthetized for 45–60 sec using 5% isoflurane and secured on the animal holder (secured with tape at the level of the thorax) of the weight-drop machine. A weight (~95 g) was dropped from 1 m high on the right side of the head to induce mild traumatic brain injury (TBI) in mice. The weight and height from where the weight would be dropped were selected based on the preliminary studies conducted at our laboratory. Group II received single induction and group III received three inductions, once daily for 3 days, while Group I underwent the same procedure except for the weight drop. Following induction, the animals were placed in the ventilator until recovery and observed for rightening reflex. The times of recovery or for the rightening reflex for Groups II and III were found to be 45–70 s. The animals were placed back in the home cage after complete recovery. The experimental design is represented in Figure 8.

### 4.4. Evaluation of Glucose Uptake in Mouse Brains Using MicroPET/CT Imaging

MicroPET/CT imaging was performed using a Vecta4CT (MILabs B. V., Ultrecht) scanner. The mice were housed under standard conditions in an approved facility. Glucose uptake was analyzed following a tail vein injection of 18–24 MBq (~20 MBq) ^18^F-FDG. Animals were anesthetized (5% isoflurane in oxygen) and injected via a tail vein with ^18^F-FDG in 0.2 mL saline. Following injection, the animals were immediately placed on the scanner bed in a supine position and secured with tape at the level of the thorax and hind limbs to prevent movement. Anesthetic was maintained using 2% isoflurane, and dynamic brain PET images were acquired for 30 min. Mice were monitored frequently for body temperature, heart rate and breathing frequency. CT images were acquired first (at 40 kV, 140-mA beam current, and 360 projections), followed by PET imaging of the whole head (30 min, 3 × 10 min frames) in a list-mode data format with a HE-GP-M-1.0 mm mouse/rat pinhole collimator. Mice were imaged following repeated injections at 24 h, and at 1, 7, 14 and 28 days post induction after which they were sacrificed. 

Acquisition of PET image was obtained along with CT and reconstruction was carried out with a pixel-based order-subset expectation maximization (POS-EM) algorithm that included resolution recovery and compensation for distance-dependent pinhole sensitivity, and quantified using the PMOD software (PMOD Technologies, Switzerland), version 3.8. ^18^F-FDG uptake in whole brain, cortex, hippocampus and cerebrum was expressed as the % injected activity per gram (%IA/cc). 

### 4.5. Motor Function Tests

#### 4.5.1. Rota Rod Test

Four chambered rota rod apparatus was used for the study. Mice were placed in the behavioral room 30 min prior to the experiment, for habituation. Mice were pre-trained in the rota rod for 3 days (three trials each day) before the test. On day 28 following last TBI induction, the mice were placed on the rod, one in each chamber facing away from the direction of rotation and then the rotation was started with an accelerated rate of 4 rpm/sec. The time on the rotating rod was recorded. The tests were repeated three times and the mean value for each animal was calculated. The maximum time and speed of the rotation were set as 120 s and 30 rpm, respectively. The apparatus was cleaned with 70% alcohol between each trial. 

#### 4.5.2. Beam Walk Test

Beam walk test was performed according to our previously published protocol [47]. Mice were pre-trained to traverse a narrow beam of 100 cm length to reach an enclosed escape platform. A bright light (approximately 60 W) was placed above the narrow beam to create an aversive stimulus. This encouraged the mice to traverse the beam to the dark enclosed goal box. On day 28 following the last TBI induction, mice were placed individually at the start of the beam and analyzed for the time taken to run across the beam, immobility periods and number of foot slips. The maximum time given for a mouse to traverse the beam was 60 s and if the mouse did not reach the goal box in 60 s, the time taken was recorded as 60 s. The beam was cleaned with 70% alcohol between each animal.

#### 4.5.3. Open Field Test

Locomotor activity was evaluated using an open field exploratory maze. On day 29, following the last TBI induction, the mice were placed at one corner of the open-field arena (40 × 40 × 30 cm; floor of the open field divided into 16 equal squares) and observed for 5 min. We recorded immobility period(s), number of total squares and center squares crossed. The floor of the maze was cleaned with 70% alcohol between each animal [48].

### 4.6. Cognitive Function-Novel Location Recognition Test

The novel location recognition test was performed to determine hippocampal-dependent cognitive function in mice [49]. The dimensions of the open arena were 40 cm length × 40 cm wide × 30 cm height. On day 29, following the last TBI induction, the mice were allowed to explore two objects, placed on the open field for a period of 15 min. After 90 min following exploration, one of the objects was placed in the novel location and the mice were tested for time spent in the novel location. The maze and the objects were cleaned with 70% alcohol between each experiment.

Following behavioral analysis, the mice were sacrificed, and brains were collected. Cortex, hippocampal and cerebellum regions were isolated and used for further analysis.

### 4.7. Measurement of Plasma PrPc Level

Plasma was separated from the blood collected from all the experimental mice for PrPc measurement, 24 h after last TBI induction. The level of PrPc in plasma was estimated using ELISA kit instruction (Biomatik Corporation, ON, Canada) and the values were expressed in mAU, as previously described [43].

### 4.8. Isolation of Lipid Raft

Cortical, hippocampal and cerebellar tissues were lysed in tissue homogenization buffer (150 mM NaCl, 20 mM Na_2_HPO_4_, 2 mM NaH_2_PO_4_, 20% (*v*/*v*) glycerol, 2 mM sodium orthovanadate with protease inhibitors) in a Dounce homogenizer and centrifuged at 10,000 rpm for 11 min at 4 °C. The supernatant was then centrifuged at 32,000 rpm for 90 min, at 4 °C to pellet the total plasma membrane. The precipitants were solubilized in 2 mL solubilizing buffer (0.5% *v*/*v* Triton X-100 in Mes-Buffered Saline (MBS; 25 mM Mes, pH 6.5/0.15 M NaCl), protease inhibitors and 2 mM sodium orthovanadate) and allowed to stand for 15 min on ice. Two mL of solubilized PM was mixed with 2 mL of 80% (*w*/*v*) sucrose in MBS buffer and applied to the bottom of a 12 mL ultracentrifuge tube. A discontinuous 5–30–40% sucrose/MBS gradient was formed by 4 mL 30% (*w*/*v*) sucrose in MBS solution on top of the 4 mL homogenate, followed by 4 mL 5% *(w/v)* sucrose in MBS solution. The sample was then centrifuged at 31,000 rpm (SW41Ti rotor) for 16 h at 4 °C to isolate the lipid raft and non-raft compartments. A light scattering band at the 5–30% interface was identified that was enriched in flotillin, indicating the presence of lipid rafts. Twelve equal fractions (1 mL each) were collected from the top of gradient to the bottom [50]. Equal volumes of each fraction were separated by SDS-PAGE and analyzed by immunoblotting as mentioned above.

### 4.9. Co-Immunoprecipitation and Immunoblotting

To detect the interaction between PrPc, flotillin and CAP, we performed co-immunoprecipitation and immunoblotting as previously described [51]. Briefly, cortical, hippocampal and cerebral slices were homogenized in ice-cold RIPA buffer (Cell Signaling, ON, Canada) and 750 µg of protein incubated with rabbit IgG and Dynabeads^®^ Protein G (Novex, MA, USA) shaken for a period of 1 h. The tube was then placed on a magnet to collect supernatant. The supernatant was added to the immunoprecipitating primary antibody (2 µg) mixture and incubated overnight at 4 °C with rotation. The beads were collected the next day and washed three times with wash buffer (1X PBS containing 100 mM NaF, 10 mM Na_2_H_2_P_2_O_7_, and 2 mM Na_3_VO_4_, 0.1% Nonidet P 40 and 0.1% Triton X-100). The proteins in the beads were eluted by adding 50 µL of 2X Laemmli sample buffer (Bio-Rad, ON, Canada) and heating at 70 °C for 10 min. The supernatant was blotted in sodium deodecyl sulfate-polyacrylamide gel electrophoresis (SDS-PAGE) and transferred onto a polyvinylidene difluoride (PVDF) membrane. The membrane was then blocked with 5% skimmed milk and incubated with primary antibody (overnight; anti-rabbit flotillin 1—1:1000 dilution, anti-mouse PrPc—1:1000 dilution or anti-mouse CAP—1:200 dilution) and HRP-conjugated secondary (clean blot, 1:4000 dilution) antibody. The membrane was washed with TBST and exposed to enhanced chemiluminescence reagent (Bio-Rad, Canada) and then imaged using Bio-Rad image analyzer. Protein bands of interest were analyzed using National Institutes of Health (NIH) ImageJ software and expressed as the relative level of target protein with that of the marker protein. 

### 4.10. Data Analysis

Statistical analysis was performed using GraphPad prism (Version 8.0). Values were expressed in mean ± standard error of the mean (SEM). For functional assessment and plasma PrPc level measurement, the mean differences between the groups were analyzed by one way ANOVA followed by Tukey’s multiple comparison test as post hoc. For quantification of glucose uptake through micro PET/CT scanning, two way ANOVA was used followed by Tukey’s multiple comparison test as post hoc; the mean differences in single and repeated TBI induction mice were compared with the respective day control group. In addition, the values of single and repeated TBI induction were compared with their respective day 0 values. A *p* value less than 0.05 was considered as statistically significant.

## Figures and Tables

**Figure 1 ijms-22-09005-f001:**
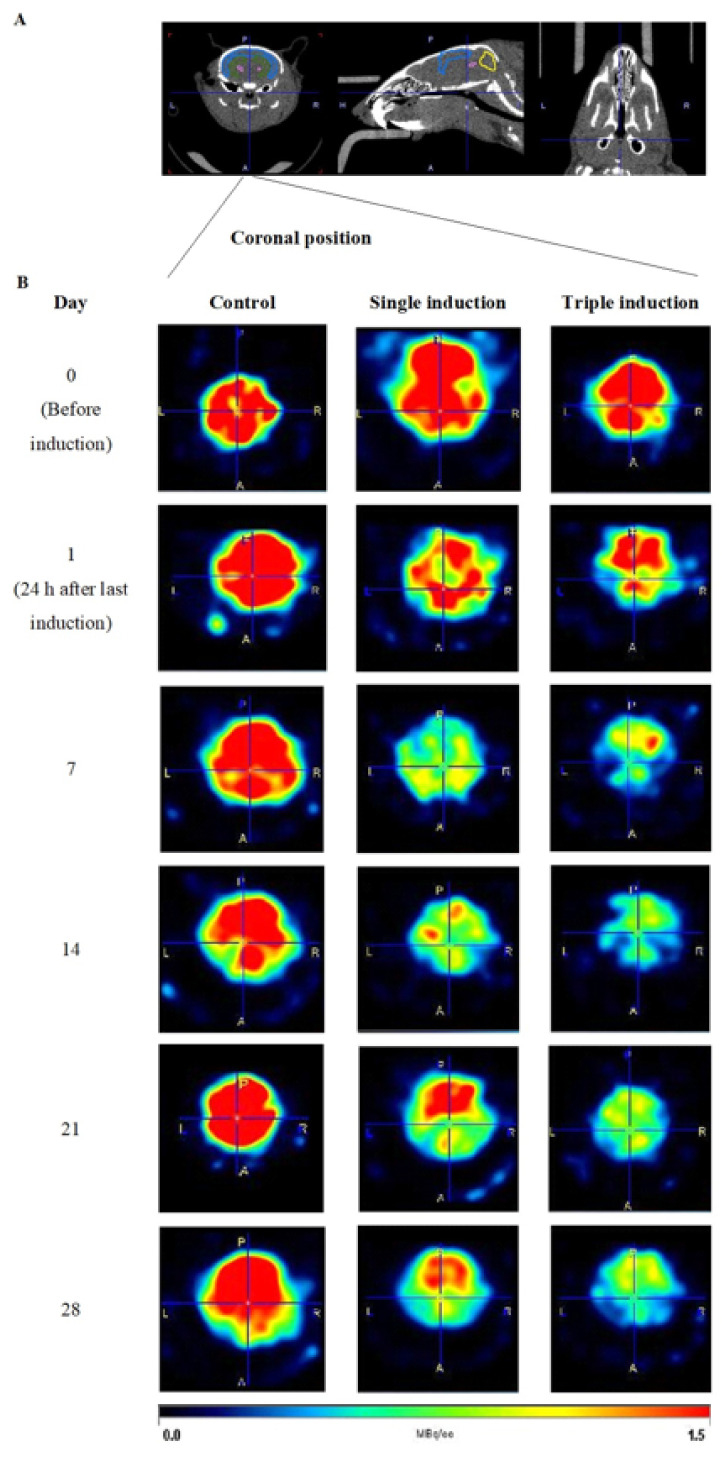
Representative micro PET images of ^18^F FDG in control, single and repeated traumatic brain injury (TBI)-induced mice brains. (**A**) Representative CT images of a mouse at coronal, horizontal and sagittal position are shown above. (**B**) PET images of control, single and repeated/triple TBI mice brains at coronal position are represented below, at days 0 (before TBI induction), 1, 7, 14, 21 and 28, after last induction. Images in the first column represent control mice brains, in the second column are single TBI-induced mice brains and the third shows repeated TBI-induced mice brains at day 0 in the first row followed by days 1, 7, 14, 21 and 28 after last induction. The intensity of red color developed is proportional to the glucose (^18^F FDG) uptake in mice brains and the scale bar is represented below.

**Figure 2 ijms-22-09005-f002:**
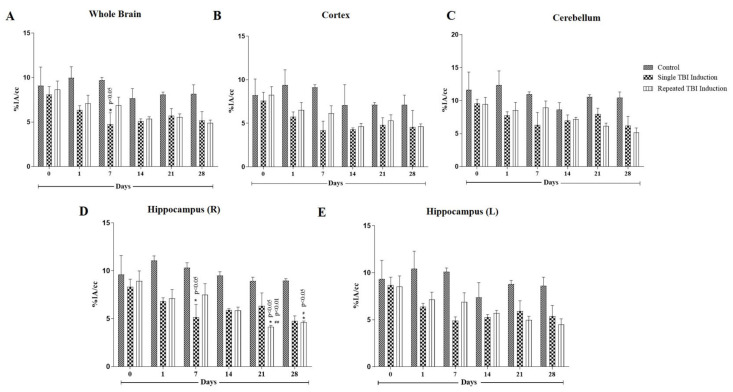
Quantification of glucose uptake in cortex, hippocampus and cerebellum regions of TBI mice brains following ^18^F-FDG injection. Glucose uptake was expressed as % injected activity per gram (%IA/cc) in whole brain (**A**), cerebrum (**B**), cerebellum (**C**) and hippocampus (**D**,**E**). ^8^F-FDG uptake was measured at days 0 (before TBI induction), 1, 7, 14, 21 and 28, after last induction. Values were expressed as mean ± SEM; *n* = 3, *n* = 3, and *n* = 4 for control, single and repeated TBI induction, respectively; mean difference between the groups were analyzed using two way ANOVA followed by Tukey’s multiple comparison test as post hoc using Graphpad prism software (v 8.0); * indicates *p* < 0.05 vs. control mice brains; #, ## indicate *p* < 0.05 and 0.01, respectively vs. glucose uptake by the respective day 0 mice brains.

**Figure 3 ijms-22-09005-f003:**
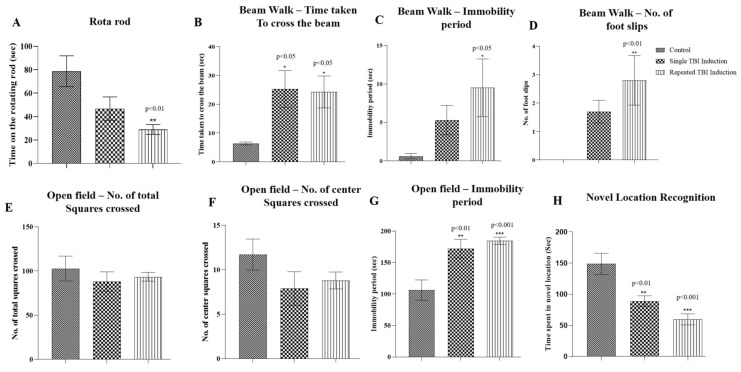
Effect of traumatic brain injury (TBI) on functional outcomes following single and repeated TBI induction in mice. Motor function was assessed using rota rod (**A**), beam walk (**B**–**D**) and open field (**E**–**G**) tests and cognitive function was assessed using novel location recognition tests in TBI mice. Bar graphs represents (**A**) time spent by the mice on the rotating rod, (**B**) time taken by the mice to cross the beam, (**C**) immobility period on the beam, (**D**) number of foot slips while walking on the beam, (**E**) number of total squares crossed in the open field, (**F**) number of center squares crossed, (**G**) immobility period in the open field and (**H**) time spent by the mice in the novel location. Values were expressed as mean ± SEM; *n* = 10; mean differences between the groups were analyzed using one way ANOVA followed by Tukey’s multiple comparison test as post hoc in Graphpad prism software (v 8.0); *, ** and *** indicate *p* < 0.05, 0.01 and 0.001, respectively vs. control mice.

**Figure 4 ijms-22-09005-f004:**
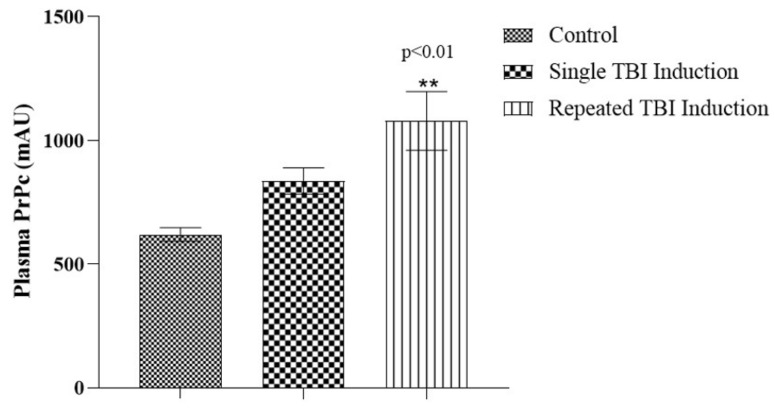
Plasma PrPc levels in TBI mice following single and repeated TBI induction. Bar diagram represents the level of PrPc in plasma (mAU); values are expressed as mean ± SEM; *n* = 6; mean differences between the groups were analyzed using one way ANOVA followed by Tukey’s multiple comparison test as post hoc in Graphpad prism software (v 8.0); ** indicate *p* < 0.01 vs. control mice.

**Figure 5 ijms-22-09005-f005:**
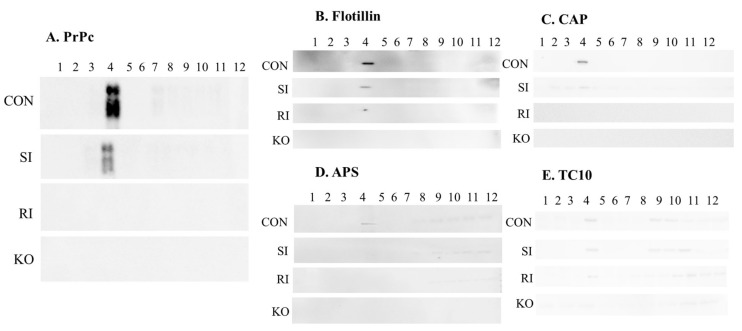
Determination of PrPc and CAP/Cbl/TC10 signaling protein levels in the lipid raft region of TBI mice brains. Levels of PrPc (**A**), flotillin (**B**), CAP (**C**), APS (**D**) and TC10 (**E**) in lipid raft regions of control, single and repeated TBI induced mice, and PrPc null mice brains were measured using lipid raft isolation followed by Western blotting technique (*n* = 3). CON—control; SI—single induction; RI—repeated induction; KO—PrPc null mice brain.

**Figure 6 ijms-22-09005-f006:**
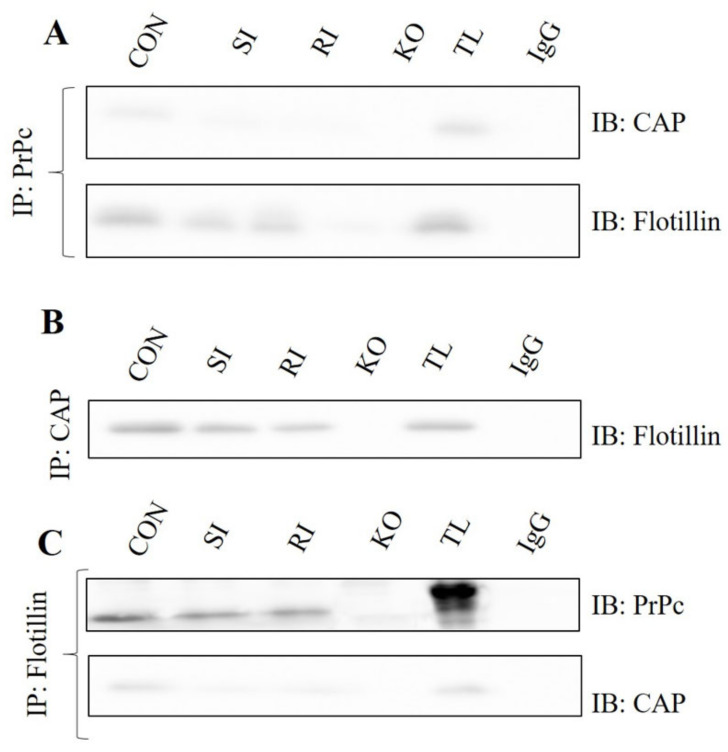
Interaction between the PrPc, flotillin and CAP were analyzed in control, single and repeated TBI mice brains using co-immunoprecipitation and Western blotting techniques. (**A**) Represents PrPc immunoprecipitated and blotted with CAP and flotillin. (**B**) Represents CAP immunoprecipitated and blotted with flotillin. (**C**) Represents flotillin immunoprecipitated and blotted with CAP and PrPc (*n* = 3). CON—control; SI—single induction; RI—repeated induction; KO—PrPc null mice brain.

**Figure 7 ijms-22-09005-f007:**
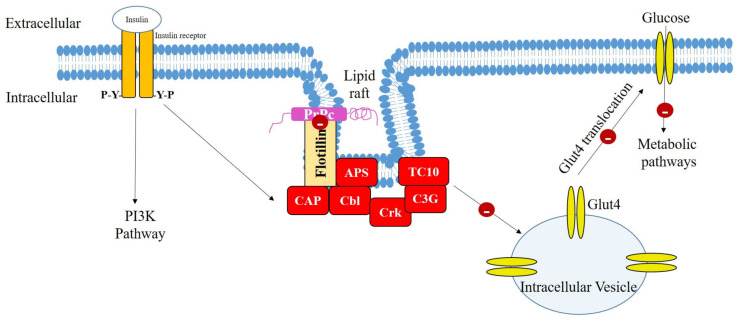
Possible mechanism of action of insulin resistance following traumatic brain injury (TBI) in mice brains. Dislodgement of PrPc results in the disruption of PrPc and flotillin interaction that may lead to the loss of CAP/Cbl/TC10 signaling cascade. This results in the inhibition of Glut4 translocation to the cell membrane and thus insulin resistance occurs. PrPc: lipid raft-localized cellular prion protein; CAP: Cbl-associated protein; APS: adaptor protein containing PH and SH2 domains; Glut4: glucose transporter 4; PI3K: phosphatidylinositol-3-kinase.

**Figure 8 ijms-22-09005-f008:**
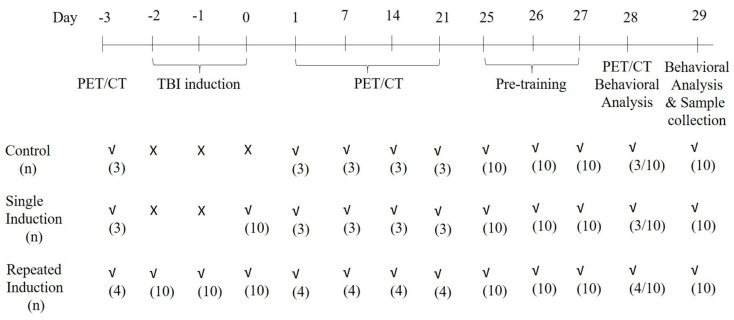
Timeline showing the TBI induction and imaging procedure for the control, single and repeated TBI-induced mice.

## Data Availability

The data presented in this study are available on request from the corresponding author.

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
