# Peer review of "Concussion/Mild Traumatic Brain Injury (TBI) Induces Brain Insulin Resistance: A Positron Emission Tomography (PET) Scanning Study"

_ijms, 2021, doi:10.3390/ijms22169005_

Round 1

Reviewer 1 Report

The manuscript of Sekar et al.  is very interesting and proposes pathophysiological considerations that could be applied to trauma patients in the future. The study is well conducted and written according to adequate scientific criteria. I don't understand why the chapters have not been placed in the traditional order (introduction, materials and methods, results, discussion); I think they need to be reorganized. Figure 1 should also be better explained in the caption. The graphs in Figure 2 and 3 are too small to display well; the images in figure 6 are too large. Finally, it would be appropriate to add further possible implications for humans to the discussion.

Author Response

Reviewer #1

The manuscript of Sekar et al.  is very interesting and proposes pathophysiological considerations that could be applied to trauma patients in the future. The study is well conducted and written according to adequate scientific criteria.

Comment 1: I don't understand why the chapters have not been placed in the traditional order (introduction, materials and methods, results, discussion); I think they need to be reorganized. Response: As per the instructions for authors, the manuscript has been aligned.

Comment 2: Figure 1 should also be better explained in the caption.

Response: As suggested by the Reviewer, figure 1 caption was explained in the revised manuscript.

Comment 3: The graphs in Figure 2 and 3 are too small to display well; the images in figure 6 are too large.

Response: As suggested by the Reviewer, the figures were customized in the revised manuscript.

Comment 4: Finally, it would be appropriate to add further possible implications for humans to the discussion.

Response: As replacing PrPc or using agents with potential upregulating effects on PrPc gene may restore the function of insulin signaling and may be beneficial as possible therapeutic agent to treat patients with TBI. Further, anti-diabetic drugs may also have therapeutic effects in the management of TBI. This implication has been included in the revised manuscript. 

Reviewer 2 Report

In the article entitled “A Potential Role for Cellular Prion Protein in Induction of  Brain Insulin Resistance Following Concussion/Mild Traumatic Brain Injury (mTBI): A Positron Emission Tomography  (PET) Imaging Study”, Sekar and colleagues evaluated links  between cellular prion protein (PrPc) and insulin resistant in a mouse model of traumatic brain injury (TBI). To study glucose uptake in TBI mice brain, mice received 18F-FDG via a tail vein following weight drop procedure. Authors compared animals submitted to the weight drop procedure once or three times, and these two groups to controls.

The article investigates an important issue, and could contribute to the field. However, I have major concerned as detailed below.

Overall, the writing should be improved, and every section of the manuscript should be revised for clarity. The lack of important information makes it difficult to evaluate the robustness of findings.

Abstract:

  1. Abstract is very general, and does not clearly explain the main findings of the manuscript.
  2. What do the authors mean by “cerebral” glucose uptake?
  3. Authors should explain how the interaction between PrPc, flotillin and Cbl associated protein (CAP) was analyzed.

Introduction:

  1. Authors should revise Introduction, providing relevant background, and better linking literature and study objectives.
  2. The term “brain injury/concussion” should not be used. The authors should instead explain that the term “concussion” is used interchangeably with mild TBI in some contexts.
  3. TBI is very heterogeneous, clinical outcomes vary with injury severity and phase (chronic vs acute). Links between TBI and diseases such as AD have been shown for more severe injuries, but this relationship is not clear for milder injuries. When reporting studies, authors fail to mention TBI severity and phase. This makes it difficult to evaluate the relevance of the provided information for the present study as well as knowledge gaps
  4. From the introduction alone, it is not clear that the authors are interested in studying mTBI.
  5. Has been shown in mTBI dysregulation of insulin-dependent glucose utilization? Please expand background information on changes in insulin metabolism after TBI, mentioning the severity of the TBI evaluated in each study.
  6. Article objectives and methods choice is not clear. For example, why comparing “single” and “triple induced” mTBI. In the discussion, authors state that ‘repeated TBI mice brain showed consistent and/or prolonged damage to insulin signaling compared to the single TBI mice brain’. This type of evidence should be presented in the Introduction to better explain the study design.
  7. It is not clear why authors measured PRPc and its relationship with insulin signaling.

Methods:

  1. Why was the weight-drop method chosen?
  2. The weight drop procedure should be better explained. For example, was the head fixated? How were the height and weight selected?
  3. Manuscript should report histological changes following weight drop procedure. Cell or tissue changes are typically looked at by using procedures such as hematoxylin and eosin, cresyl violet, or Nissl staining.
  4. The number of animals is (n = 10) is rather small. How was the necessary number of animals determined? How many animals were used in each analysis? Figure 4 reports a n of 6 animals. Why the discrepancy? What is the number of animals in each experimental group?
  5. How long after the weight drop procedures was the imaging undertaken? Are the number of days counted since the first or last induction?
  6. When was each of the behavioral tests performed? Please describe timepoints explaining if they refer to first or last induction. If performed in the same day, were the behavioral tests performed before or after the imaging procedure?
  7. Timeline for procedures should be better explained in the article. A figure showing the timeline as well as well as number of animals in each procedure/per group would help.
  8. Please better explain the statistical analysis performed in the manuscript, including levels of each variable.

Results:

  1. Lines 84-94: Changes in glucose uptake are described very vaguely. How much is a “marked decreased”?
  2. Authors report that “The data showed that repeated TBI induction results in permanent decrease in glucose metabolism in whole brain, cerebrum, cerebellum and hippocampal regions in mice”. Is this decrease statistically significant when repeated TBI is compared to “single induction” and control animals? Please provide p values.
  3. Authors use the terms ‘repeated TBI’ as well as ‘triple induction’ in the manuscript. Uniformity in terminology would improve the manuscript clarity. Please consider using the term single and repeated (or repetitive or multiple) TBIs instead of single or triple induction throughout the manuscript .
  4. For each comparison, provide p values for ANOVA and posthoc analysis, highlighting statically significant findings.
  5. Please add statically significant p values and asterisks to bar graphs (showing statically significant pairwise comparison).

Discussion:

  1. Do authors believe their findings and conclusions are sound considering the small number of animals?
  2. Please discuss limitations of the present study.
  3. In this study, only male animals were used. Please add this to limitations. Have been reported sex differences in insulin resistance post-TBI?
  4. Please add references to support statements.

Author Response

Reviewer #2 (Remarks to the Author):

In the article entitled “A Potential Role for Cellular Prion Protein in Induction of Brain Insulin Resistance Following Concussion/Mild Traumatic Brain Injury (mTBI): A Positron Emission Tomography (PET) Imaging Study”, Sekar and colleagues evaluated links between cellular prion protein (PrPc) and insulin resistant in a mouse model of traumatic brain injury (TBI). To study glucose uptake in TBI mice brain, mice received 18F-FDG via a tail vein following weight drop procedure. Authors compared animals submitted to the weight drop procedure once or three times, and these two groups to controls.

The article investigates an important issue, and could contribute to the field. However, I have major concerned as detailed below.

Overall, the writing should be improved, and every section of the manuscript should be revised for clarity. The lack of important information makes it difficult to evaluate the robustness of findings.

Abstract:

Comment 1: Abstract is very general, and does not clearly explain the main findings of the manuscript.

Response: As suggested by the Reviewer, the abstract has been revised.

Comment 2: What do the authors mean by “cerebral” glucose uptake?

Response: We apologize for the typographical error. It was cerebellum and the same has been modified in the revised manuscript.

Comment 3: Authors should explain how the interaction between PrPc, flotillin and Cbl associated protein (CAP) was analyzed.

Response: Thank you for the comment. As suggested by the Reviewer 2, the abstract has been modified in the revised manuscript.

Introduction:

Comment 4: Authors should revise Introduction, providing relevant background, and better linking literature and study objectives.

Response: As suggested by the Reviewers, the Introduction has been modified accordingly.

Comment 5: The term “brain injury/concussion” should not be used. The authors should instead explain that the term “concussion” is used interchangeably with mild TBI in some contexts.

Response: As suggested by the Reviewer, we corrected the introduction accordingly.

Comment 6: TBI is very heterogeneous, clinical outcomes vary with injury severity and phase (chronic vs acute). Links between TBI and diseases such as AD have been shown for more severe injuries, but this relationship is not clear for milder injuries. When reporting studies, authors fail to mention TBI severity and phase. This makes it difficult to evaluate the relevance of the provided information for the present study as well as knowledge gaps

Response: We agree with the respected reviewer on the heterogenous nature of TBI events and dependence of clinical outcome with severity of brain injury. We also agree that developing neurodegenerative diseases including AD is relevant with chronic (repeated) severe cases of TBI such as those cases with chronic severe sport mTBI (professional boxers, football players, etc.).  The point of respected reviewer is well taken by the authors that our animal model of repeated mTBI may not exactly reflect the real-life cases of developing neurodegenerative diseases such as AD in chronic severe TBIs, but due to ethical limitations of working with laboratory animals for mTBI, this was the closest model that the authors were able to come to.    In this study our main purpose was to highlight the potential involvement of PrPc in pathogenesis of mTBI which warrants more future studies.

As mentioned in our manuscript an association between the repeated mild TBI and insulin resistance has been demonstrated in various studies. Further, it is well known that persistent insulin resistance leads to type 2 diabetes mellitus if unattended. Since the mild TBI patients are unattended mostly, they may develop brain insulin resistance and long-term neurodegenerative diseases. Thus we selected mild TBI mouse model to study the molecular link in the generation of insulin resistance in mice. As we propose the molecular link between PrPc and CAP, disruption of interaction may lead to insulin resistance.  

Comment 7: From the introduction alone, it is not clear that the authors are interested in studying mTBI.

Response: The Introduction in the revised manuscript has been modified accordingly.

Comment 8: Has been shown in mTBI dysregulation of insulin-dependent glucose utilization? Please expand background information on changes in insulin metabolism after TBI, mentioning the severity of the TBI evaluated in each study.

Response: Perkind and his team in 2011 demonstrated that repeated mild TBI patients showed decreased glucose metabolism which was studied in PET imaging. However, the molecular link between them has not been established yet. The present study was made an attempt to investigate the interaction between them and we observed that PrPc may play a molecular link between them.

Comment 9: Article objectives and methods choice is not clear. For example, why comparing “single” and “triple induced” mTBI. In the discussion, authors state that ‘repeated TBI mice brain showed consistent and/or prolonged damage to insulin signaling compared to the single TBI mice brain’. This type of evidence should be presented in the Introduction to better explain the study design.

Response: we have clear evidences to support that repeated TBI generates disruption of insulin signaling cascade in patients and animal models. Further, we previously showed the decreased PrPc level in single TBI induction in mice model. Based on this, we attempted to study the effect in single and repeated inductions and compare which model generates clear and evident impairment in glucose utilization. We observed that repeated induction showed persistent reduction of glucose utilization.

Comment 10: It is not clear why authors measured PRPc and its relationship with insulin signaling.

Response: In our previous experiments, we published that PrPc acts as a biomarker for TBI. Dislodgment of PrPc from the cell membrane to the circulation was observed in sports concussion patients and mild TBI mouse model. There are evidences for flotillin and CAP interaction in lipid raft region which is associated with the insulin signaling processes. Also, it is evident that PrPc has interaction with flotillin and involved in various physiological functions. We proposed that PrPc may have interaction with CAP directly or indirectly through flotillin. Thus dislodgement of PrPc may lead to insulin resistance.

Methods:

Comment 11: Why was the weight-drop method chosen?

Response: The reason why we chosen weight-drop method is that the injury developed will be mild while the majority of the present TBI techniques impose such intense injuries. Further, this method is easy for studying repeated injury in mice model. In addition, the injury through weight-drop method generates a more clinically relevant concussive-like symptomology. We also had previous publications with this model. Thus we chosen this method.

Comment 12: The weight drop procedure should be better explained. For example, was the head fixated? How were the height and weight selected?

Response: As suggested by the Reviewer, the manuscript has been revised accordingly.

Comment 13: Manuscript should report histological changes following weight drop procedure. Cell or tissue changes are typically looked at by using procedures such as hematoxylin and eosin, cresyl violet, or Nissl staining.

Response: We have performed immunohistochemistry for NeuN staining (marker for neuronal cells) for control and repeated TBI mice brain in cortical and hippocampal regions. We found a significant decrease in NeuN staining in repeated TBI mice brain compared to the control mice. The data was published in the Journal of Neurotrauma in 2019. However, we will consider the Reviewer’s suggestion in our future investigations.

Comment 14: The number of animals is (n = 10) is rather small. How was the necessary number of animals determined? How many animals were used in each analysis? Figure 4 reports a n of 6 animals. Why the discrepancy? What is the number of animals in each experimental group?

Response: n=10 will be the minimum number of animals used generally for statistical analysis in case of behavioral analysis. However, we agree with the Reviewer and it will be included in the conclusion section as a limitation. Further, for behavioral analysis, we used 10 animals per group; we collected blood randomly from 6 animals per group, that is why we mentioned n=6. For co-immunoprecipitation and lipid raft isolation experiments, we used 3 animals/group. For PET scanning, 3 animals for control and single TBI and 4 animals for repeated TBI were used. 

Comment 15: How long after the weight drop procedures was the imaging undertaken? Are the number of days counted since the first or last induction?

Response: We performed imaging once before TBI induction. Then we induced TBI (both single and repeated). 24 h following last day of induction (next day in case of single induction and day 4 in case of repeated induction), the day 1 imaging was carried out. Following this, the imaging was performed on day 7, 14, 21 and 28 for each animal.

Comment 16: When was each of the behavioral tests performed? Please describe time points explaining if they refer to first or last induction. If performed in the same day, were the behavioral tests performed before or after the imaging procedure?

Response: The pre-training for rota rod and beam walk tests were performed on day 25, 26 and 27 after last induction. The test for rota rod and beam walk was performed on day 28, and for open field and novel location recognition on day 29. The behavioral tests were performed 2 hours after the imaging procedure. The details have been included in the revised manuscript.

Comment 17: Timeline for procedures should be better explained in the article. A figure showing the timeline as well as well as number of animals in each procedure/per group would help.

Response: As suggested by the Reviewer, a figure for the timeline was added in the revised manuscript.

Comment 18: Please better explain the statistical analysis performed in the manuscript, including levels of each variable.

Response: As suggested by the Reviewer, the statistical analysis in method section has been modified in the revised manuscript.

Results:

Comment 19: Lines 84-94: Changes in glucose uptake are described very vaguely. How much is a “marked decreased”?

Response: The term marked decrease was used to describe the changes happened in figure 1, between control and single; control and repeated induction. However, the statistical analysis for the quantification was explained in the next paragraph. As many of the data does not attain the statistical significance value due to high SEM, we calculated the percentage decrease between them and the values were included in the revised manuscript.

Comment 20: Authors report that “The data showed that repeated TBI induction results in permanent decrease in glucose metabolism in whole brain, cerebrum, cerebellum and hippocampal regions in mice”. Is this decrease statistically significant when repeated TBI is compared to “single induction” and control animals? Please provide p values.

Response: We thank Reviewer for this comment. Since we did not attain statistical significance values for all the brain regions, we modified as “the images” in the revised manuscript. Also, increased animal numbers may attain a clear statistical values and the statement was included in the revised manuscript.

Comment 21: Authors use the terms ‘repeated TBI’ as well as ‘triple induction’ in the manuscript. Uniformity in terminology would improve the manuscript clarity. Please consider using the term single and repeated (or repetitive or multiple) TBIs instead of single or triple induction throughout the manuscript.

Response: As suggested by the Reviewer, the manuscript has been modified accordingly.

Comment 22: For each comparison, provide p values for ANOVA and posthoc analysis, highlighting statically significant findings.

Response: If we attain the significant level when comparing between groups, we included in the manuscript and if we did not observed statistically significant level, we have note mentioned in the manuscript.

Comment 23: Please add statically significant p values and asterisks to bar graphs (showing statically significant pairwise comparison).

Response: As suggested by the Reviewer, the figures have been modified in the revised manuscript.

Discussion:

Comment 24: Do authors believe their findings and conclusions are sound considering the small number of animals?

Response: As commended by the Reviewer, this statement is included in the revised manuscript.

Comment 25: Please discuss limitations of the present study.

Response: The limitations in the present study was added in the discussion part of the revised manuscript.

Comment 26: In this study, only male animals were used. Please add this to limitations. Have been reported sex differences in insulin resistance post-TBI?

Response: Various studies were performed in both sexes and the results obtained was highly variable. However, female animals showed protective effect in behavioral outcomes and pathology following TBI. Thus we used male mice in the present study and it has been included in the discussion part of the revised manuscript.

Comment 27: Please add references to support statements.

Response: Appropriate references have been included in the revised manuscript.